# A New Synthesis Strategy on Styrene-Butadiene Di-Block Copolymer Containing High *cis*-1,4 Unit Via Transfer of Anionic to Coordination Polymerization

**DOI:** 10.3390/polym11020195

**Published:** 2019-01-23

**Authors:** Jie Liu, Xin Min, Xiuzhong Zhu, Zichao Wang, Tong Wang, Xiaodong Fan

**Affiliations:** 1Ministry of Education and Shaanxi Key Laboratory of Macromolecular Science and Technology, School of Science, Northwestern Polytechnical University, Xi’an 710072, China; liujie509_1982@126.com (J.L.); 15991672082@163.com (X.M.); zhuxiuzhong321@163.com (X.Z.); wc19910102@126.com (Z.W.); 17795837806@163.com (T.W.); 2School of material scienece and Engineering, Shaanxi University of Technology, Hanzhong 723001, China

**Keywords:** styrene-butadiene di-block copolymer, high *cis*-1,4 unit, anionic polymerization, coordination polymerization, mechanistic transfer

## Abstract

A novel synthesis strategy on styrene-butadiene di-block copolymer (PS-*b*-PB) with high *cis*-1,4 unit content was developed, based on a transfer technique from anionic to coordination polymerization. Firstly, the styrene monomer was initiated by *n*-butyllithium (Li) utilizing anionic polymerization at 50 °C, which resulted in a macromolecular alkylating initiator (PSLi). Secondly, PSLi was aged with nickel naphthenate (Ni) and boron trifluoride etherate (B) for obtaining a complex catalyst system (Ni/PSLi/B). Then, Ni/PSLi/B was applied to initiate the butadiene (Bd) polymerization. Following this new strategy, a series of PS-*b*-PBs were successfully synthesized. The experimental results indicated that under the molar ratio combination of [Li]/[Ni] = 5 and [B]/[Li] = 1, styrene-butadiene di-block copolymers could be easily achieved with high *cis*-1,4 unit content (>97%) and controlled molecular weight as well as narrow molecular weight distribution (M_w_/M_n_ < 1.5). Furthermore, the copolymer’s block ratio could also be effectively controlled by controlling the two components’ monomer feed ratio.

## 1. Introduction

Styrene-butadiene di-block copolymers (PS-*b*-PB) usually consist of a soft segment, polybutadiene (PB), and a hard segment, polystyrene (PS). These copolymers offer excellent mechanical and thermal properties, making them fulfill wide industrious applications [1,2,3,4,5]. To date, various synthesis strategies on styrene-butadiene block copolymers were achieved including radical and anionic polymerizations. Radical polymerization, such as ATRP [6] and RAFT [7], has especially been used to prepare di- and multi-block copolymers with styrene and butadiene as the monomers. However, the stereo-structure in these copolymers cannot be effectively controlled. The anionic polymerization utilizing alkyl-lithium as the initiator is widely used in the synthesis of styrene-butadiene block copolymers. Although the molecular weight distribution using this technique is narrow, the product’s stereo-structure still could not be sufficiently controlled, whereas the *cis*-1,4 unit content only reached about 35–40 mol% [8]. According to the current theory, the high content of the *cis*-1,4 unit not only enhances the copolymer’s mechanical properties at much lower temperature, but also effectively changes the tensile strength and flexibility of the styrene-butadiene copolymers products [9]. Unfortunately, radical and anionic polymerizations still exhibit certain limitations for effectively controlling the stereo-structures of PBs, leading to obtaining copolymers with higher *cis*-1,4 unit content [10,11].

It is well-known that coordination polymerization is a powerful synthesis approach to control the regulated stereo-structures of polydienes. For example, the styrene or butadiene polymerization can be initiated via coordination polymerization. A homopolybutadiene with a content of *cis*-1,4 unit as high as 98% [12,13,14], and a homopolystyrene with high degree of tacticity structure [15,16,17] can be respectively obtained. However, due to the difference in catalytic activity of the same coordination catalyst system for styrene and butadiene monomers, the precise control on the content of these monomers appears to be very difficult [1,18]. Hence, recently, a number of research groups have focused on the development of styrene-butadiene di-block and tri-block copolymers using complex coordination polymerization systems. For instance, Zhu et al. have synthesized styrene-butadiene di-block copolymers with high *cis*-1,4 content (90–96%) using a ternary neodymium catalyst [19]. Nevertheless, after completing the polymerization of styrene monomer, the conversion of butadiene in the second step was only about 50%, and the obtained copolymer also exhibited a wide molecular weight distribution. Zambelli [20] and Ban [21] have reported that stereo-regular styrene-butadiene tri-block copolymers (SBS) could be fairly synthesized using complex metallocene-titanium with methylaluminoxane (MAO), but the conversion of butadiene in the second step was also very low. Cui et al. [10] synthesized PS-*b*-PB with high *cis*-1,4 unit content (*cis*-1,4 ~ 95%) by utilizing catalyst [C_5_Me_4_-C_5_H_4_N)Lu(*η*^3^-C_3_H_5_)_2_]. Hou et al. [22,23] also have synthesized the PS-*b*-PB with 99% *cis*-1,4 unit content using a Half-Sandwich samarium catalyst. However, the complicated and expensive preparation process of this Half-Sandwich rare earth catalyst and the cost of the co-catalyst [Ph_3_C][B(C_6_F_5_)_4_] seriously limited their industrial-scaled synthesis and application.

Based on the problems mentioned above, the present study aims to synthesize a styrene-butadiene di-block copolymer (PS-*b*-PB) with high *cis*-1,4 unit content via a combination of anionic polymerization with coordination polymerization. The synthesis routes are schematically illustrated in Scheme 1.

Firstly, *n*-butyllithium (Li) was utilized to initiate styrene polymerization for obtaining a macromolecular alkylating initiator (PSLi). Then, PSLi was reacted with nickel naphthenate (Ni) and boron trifluoride etherate (B) to obtain a complex catalyst system (Ni/PSLi/B), which was further used for butadiene polymerization. The resulted PS-*b*-PB not only possessed a high *cis*-1,4 unit content, but also had a narrow molecular weight distribution. The detailed synthesis procedures suggested that the combination of anionic with coordination polymerization was a simple and efficient strategy to synthesize diene based di-block copolymers with a high *cis*-1,4 unit content, which could lead to excellent mechanical properties at low temperatures.

## 2. Experimental

### 2.1. Materials

The 1,3-butadiene (Bd, 1.9 M in *n*-hexane) was obtained from Energy China (Energy Chemical Technology Co., Ltd., Shanghai, China) and used as received. Styrene (St) purchased from Macklin China (Macklin Biochemical Co., Ltd., Shanghai, China) was purified by distillation at a reduced pressure over calcium hydride and stored at −18 °C with dilution to 2 M in cyclohexane. Nickel (II) naphthenate (Ni, 5 wt %) supplied by Meryer China was diluted to 0.025 M with cyclohexane, and *n*-butyllithium (Li, 2.5 M in *n*-hexane) was obtained from Energy China. Boron trifluoride etherate (B, 98%) and 2,6-di-tert-butyl-p-cresol (antiager 264, CP) were purchased from Macklin (Macklin Biochemical Co., Ltd., Shanghai, China). Cyclohexane was distilled with calcium hydride and stored in a 4 Å molecular sieve for one week before use. Methanol (AR), ethanol (AR) and methyl ethyl ketone (AR) were received and used without further purification.

### 2.2. Synthesis of PS-b-PB via Ni/PSLi/B Catalyst System

In the first step, the polymerization of styrene was carried out under anionic polymerization mechanism. The initiator *n*-butyllithium was used directly as purchased. The obtained PSLi was used to prepare Ni/PSLi/B complex catalyst. All syntheses were conducted in a dry argon atmosphere. A detailed polymerization procedure (PS-*b*-PB-4, provided in Table 1) was described as a typical example. First, styrene polymerization was conducted in Schlenk tube with a rubber septum in following steps: styrene solution (10 mmol, 2 M in cyclohexane, [St]/[Li] = 200), Li (0.05 mmol, 2.5 M in *n*-hexane, [Li]/[Ni] = 5), the polymerization was carried out at 50 °C for 1 h, and viscous orange liquid was obtained. Then the nickel naphthenate solution (Ni, 0.01 mmol, 0.025 M in cyclohexane) was added, the system turned its color to deep red under continuous stirring at 50 °C. After 15 min, boron trifluoride etherate (B, 0.05 mmol, [B]/[Li] = 1) was added, and aged with stirring at 50 °C for 15 min. Then a dark brown-colored catalyst Ni/PSLi/B, was successfully obtained. For the last step, butadiene (Bd, 10 mmol, 1.9 M solution in *n*-hexane, [Bd]/[Ni] = 1000) and the aforementioned prepared catalyst Ni/PSLi/B solution were injected into the Schlenk tube. This polymerization was carried out at 50 °C for 3 h and then quenched by adding 2 mL ethanol containing antiager 264 (1 wt %) as a stabilizer. The product was precipitated in methanol and repeatedly washed with ethanol, followed by extraction with methyl ethyl ketone and *n*-hexane, respectively. Then, the product was dried under vacuum at 40 °C for getting a constant weight. Finally, a white solid was obtained. A series of other copolymers, such as PS-*b*-PB-1, PS-*b*-PB-2, PS-*b*-PB-3, PS-*b*-PB-5, PS-*b*-PB-6 (as shown in Table 2 for details), were synthesized by changing the amount of butadiene added in the second step of the polymerization with above synthesis process.

### 2.3. Synthesis PS-b-PB via n-Butyllithium Catalyst System

All syntheses were conducted in a dry argon atmosphere. The specific experimental process (PS-*b*-PB-a, provided in Table 1) is as follows. First, styrene polymerization was conducted in Schlenk tube with a rubber septum in following steps: styrene solution (15 mmol, 2 M in cyclohexane, [St]/[Li] = 300), Li (0.05 mmol, 2.5 M in *n*-hexane), the polymerization was carried out at 50 °C for 1 h, and viscous orange liquid was obtained. Then butadiene (Bd, 50 mmol, 1.9 M solution in hexane, [Bd]/[Li] = 1000) was injected into the Schlenk tube. This polymerization was carried out at 50 °C for 12 h, and then quenched by adding 2 mL ethanol containing antiager 264 (1 wt %) as a stabilizer. The product was precipitated in methanol and repeatedly washed with ethanol, followed by extraction with methyl ethyl ketone and *n*-hexane, respectively. Then the product was dried under vacuum at 40 °C to achieve a constant weight. Finally, a white solid was obtained.

### 2.4. Homopolymerization of Butadiene

All syntheses were conducted in a dry argon atmosphere. The specific experimental process (PB-1, provided in Table 1) is as follows. First, the alkylation process of nickel naphthenate (Ni) was conducted in a Schlenk tube with a rubber septum in the following steps: Ni (0.01 mmol, 0.025 M in cyclohexane), Li (0.05 mmol, 2.5 M in *n*-hexane, [Li]/[Ni] = 5). The reaction was carried out at 50 °C for 15 min, then the B (0.05 mmol, [B]/[Li] = 1) was added by a micro-injector and aged with stirring at 50 °C for 15 min, and a dark brown-colored catalyst Ni/Li/B, was successfully obtained. Then butadiene (10 mmol, 1.9 M solution in hexane, [Bd]/[Ni] = 1000) and the aforementioned prepared catalyst Ni/Li/B solution were injected into the Schlenk tube. This polymerization was carried out at 50 °C for 3 h and then quenched by adding 2 mL ethanol containing antiager 264 (1 wt %). The product was precipitated in methanol and repeatedly washed with ethanol. Then, the product was dried under vacuum at 40 °C to achieve a constant weight. Finally, a white solid was obtained.

### 2.5. Molecular Structure Characterizations

The macromolecular structures of the copolymers were analyzed by ^1^H NMR and ^13^C NMR spectroscopy (Bruker, 400 MHz, Madison, WI, USA) in CDCl_3_ using tetramethylsilane as an internal standard. The actual ratio of the components of PS and PB segments in copolymer was estimated by ^1^H NMR, according to the previously published method [10]. In addition, the ratio of 1,4- and 3,4-structure content was also determined by ^1^H NMR, and the ratio of *cis*-1,4 and *trans*-1,4 structure content was determined by ^13^C NMR [10,19].

FT-IR spectra were recorded using a Bruker Nicolet iS 10 spectrophotometer (Madison, WI, USA).

The number-average molecular weight (M_n_) and polydispersity indices (M_w_/M_n_) of the PS-*b*-PBs were measured by a DAWN EOS size exclusion chromatography/multi-angle light scatter instrument (SEC-MALLS, Wyatt Technology, Santa Barbara, CA, USA). HPLC grade THF was used as an eluent with a flow rate of 0.5 mL min^−1^ at 25 °C.

The glass transition temperatures of copolymers were measured by using differential scanning calorimetry (DSC 200 PC, Netzsch Instruments, Selb, Germany). The 10 mg sample was scanned at a scan rate of 10 °C/min from −130 °C to 200 °C. To avoid the effect of thermal hysteresis, the sample was first heated to 200 °C and cooled down to −130 °C, followed by DSC scanning in a settled temperature range.

## 3. Results and Discussion

### 3.1. Confirmation of Resulted Copolymer Structures

The FT-IR, ^1^H NMR and ^13^C NMR spectra of the PS-*b*-PB prepared via two steps with Ni/PSLi/B systems are presented in Figure 1, Figure 2 and Figure 3, respectively. The FT-IR spectrum exhibited two distinct absorption bands at 697 cm^−1^ and 1500 cm^−1^, corresponding to the out-of-plane deformation vibrations and skeleton stretching vibrations of the benzene ring in the PS-*b*-PB. The bands at 745 cm^−1^, 967 cm^−1^ and 911 cm^−1^ represent the characteristic absorption peaks of *cis*-1,4, *trans*-1,4 and 1,2-structures in the butadiene segment. These data confirmed that PS-*b*-PB had been successfully synthesized via the Ni/PSLi/B catalyst systems. As per previous reports [12], the data revealed that the di-block copolymer has very high *cis*-1,4 PB unit content (*cis*-1,4 > 97%). As shown in the Figure 2, the chemical shifts at 5.45 ppm and 5.05 ppm correspond to the 1,4- and 1,2-structures of the PB segment, whereas the chemical shifts between 6.5 to 7.2 ppm represent the benzene ring in PS segment, and as presenting in Figure 3. The chemical shifts at 27.5 ppm, 32.8 ppm, 34.3 ppm and 145.3 ppm represent the *cis*-1,4, *trans*-1,4 and 1,2- and PS, respectively. The integration of corresponding chemical shifts indicated that the PS-*b*-PB polymer contained a relatively higher *cis*-1,4 unit content [10,19].

The size exclusion chromatography (SEC) curves of samples of the PS (obtained in first step) and PS-*b*-PB (obtained in the second step) were shown in Figure 4, the curve of PS-*b*-PB shifted to the higher molecular weight regions with the addition of butadiene monomer while the unimodal shape was maintained. This result reflected that there exist no homopolymers in the PS-*b*-PB copolymer.

### 3.2. Comparison of Ni/PSLi/B and n-Butyllithium Catalyst Systems

As the *n*-butyllithium anionic catalyst possessing higher catalytic activity for styrene, the macromolecular alkylating initiator, PSLi, was firslty synthesized by anionic polymerization. Afterwards, PSLi was reacted with nickel naphthenate (Ni) and boron trifluoride diethyl ether (B) for obtaining a nickel-based Ni/PSLi/B catalyst system. The complex (Ni/PSLi/B) catalyst was then used to initiate butadiene polymerization for PS-*b*-PB copolymer. In order to make a comparison, the PS-*b*-PB was also synthesized only by *n*-butyllithium initiator based anionic polymerization. The results are summarized in Table 1.

Table 1 shows that the content of *cis*-1,4 (*cis*-1,4 = 97.1%) in the PS-*b*-PB obtained via Ni/PSLi/B catalyst system (Ni/PSLi/B system) is much higher than that of the copolymer (*cis*-1,4 = 48.9%) via *n*-butyllithium only. The results imply that the combination of anionic with coordination polymerization indeed has the capacity to fulfill stereo-structure controlling, especially in terms of the use of the nickel-based catalyst. Moreover, the PS-*b*-PB copolymer via Ni/PSLi/B system also presented a narrow molecular weight distribution (M_w_/M_n_ = 1.29).

The comparisons of FT-IR and ^1^H NMR spectra of the PS-*b*-PBs, obtained by different catalyst systems, are presented in Figure 5 and Figure 6, respectively. Although both catalyst systems demonstrated the effect of successful syntheses of PS-*b*-PB copolymers, Ni/PSLi/B system particularly exhibited an ability to achieve a much higher *cis*-1,4 unit content (>97%). 

Furthermore, the absorption peak of the butadiene segment at 5.05 ppm via *n*-butyllithium catalyst was significantly stronger than that of the Ni/PSLi/B system, indicating the content of 1,2-unit structure in this copolymer is very high (Figure 6).

Additionally, in order to show the anion to coordination conversion process during the polymerization process, the catalyst system Ni/Li/B, obtained by aging the Nickel (II) naphthenate (Ni) with *n*-butyllithium(Li) and boron trifluoride ether (B), was studied to directly initiate the homopolymerization of butadiene (PB-1, as shown in Table 1). The results revealed that the homopolymer of PB also possesses high *cis*-1,4 unit content (*cis*-1,4 = 96.7%), which was similar to its stereo-structure in the PS-*b*-PB. This indicated that the chain growth of butadiene segments with this complex catalyst system was also following the coordination polymerization mechanism [24]. In accordance with the literature [25,26,27], the proposed mechanism of polymerization of butadiene can be drawn as in Scheme 2. 

Due to the coordination polymerization mechanism, Ni/PSLi/B catalyst system could not only be efficient for initiating the polymerization of butadiene, but also had a stereo-selectivity under the nickel-based catalyst to achieve a high *cis*-1,4 unit content (>97%) in PS-*b*-PB. However, PSLi notably behaved as an alkylating agent only and did not play the initiator role during butadiene polymerization.

### 3.3. The Determination of the Ratio of [Li]/[Ni] and [B]/[Li]

It was found that the complex catalyst system including each individual catalyst’s molar ratio such as *n*-butyllithium versus nickel naphthenate ([Li]/[Ni)), boron trifluoride diethyl ether versus *n*-butyllithium ([B]/[Li]) could seriously affect the monomer’s conversion on PS-*b*-PBs. The result is shown in Figure 7. When [Li]/[Ni] molar ratio was less than 2.5, the conversion of butadiene was very low. It could be subject to incomplete alkylation of nickel naphthenate (Ni). With the increase in amount of *n*-butyllithium, the reactivity of butadiene gradually increased. Figure 7b shows that without boron trifluoride etherate, the Ni/PSLi/B system presented no catalytic activity on butadiene. However, the excess of boron ([B]/[Ni] > 2) could then decrease the content of *cis*-1,4 units. Therefore, an optimal ratio in which the precise amount of [Li] and [B] was determined could guarantee an efficient polymerization. Our exhausting experimental data indicated that the molar ratio of [Li]/[Ni] = 5 and [B]/[Li] = 1 could completely ensure the conversion of butadiene, reaching more than 91%, and that PS-*b*-PB has a high *cis*-1,4 unit content with a narrow molecular weight distribution (M_w_/M_n_ = 1.29).

### 3.4. The Effect of the Monomer Feed Ratio of [Bd]/[St] on the Structure of PS-b-PB

Table 2 presents the macromolecular characteristics of a series of PS-*b*-PBs via Ni/PSLi/B catalyst system with molar feed ratio of butadiene verse styrene monomer. By changing the amount of butadiene under the styrene fixed to 10 mmol, the effect of [Bd]/[St] feed ratio on copolymer’s structure was investigated via a systematic polymerization process. A clearly seen from Table 2, the actual percentage of the polystyrene component in copolymer could be effectively controlled via the monomer feed ratio of [Bd]/[St].

In order to confirm the actual effect of copolymer’s structure character with higher *cis*-1,4 unit content on its physical properties, the thermal behaviors of PS-*b*-PBs synthesized via two different catalyst systems including *n*-butyllithium and Ni/PSLi/B, were measured by DSC, as shown in Figure 8. Evidently, PS-*b*-PB via Ni/PSLi/B in which polystyrene component is 34.5% (Sample PS-*b*-PB-3, in Table 2), presented a glass transition temperature, *T_g_* at −103.2 °C, and PS-*b*-PB synthesized via *n*-butyllithium system exhibited *T_g_* at −81.6 °C (Sample PS-*b*-PB-a, in Table 1). The results indicated that the copolymer’s stereo-structure can significantly determine their related physical properties. The copolymer with higher *cis*-1,4 unit content which inherently possesses a much lower molecular internal rotation energy can naturally exhibit a much lower *T_g_*. This feature could maintain a nice flexibility at an extremely low temperature. On the other hand, the PS-*b*-PB synthesized via *n*-butyllithium system has *cis*-1,4 unit content only to 48.9%. Namely, its 1,2-unit content in a whole copolymer chain could account for as high as 11%. As the glass transition temperature for 1,2-unit segments are usually around −15 °C, it was therefore predictable that copolymer via *n*-butyllithium must have relatively poor mechanical properties at the lower temperature.

## 4. Conclusions

In this paper, a macromolecular alkylating initiator (PSLi) which was obtained by the reaction of styrene and *n*-butyllithium was aged with nickel naphthenate (Ni) and boron trifluoride etherate (B) to further prepare a complex catalyst system, Ni/PSLi/B. This catalyst, Ni/PSLi/B, then was employed to initiate the polymerization of butadiene to form a di-block copolymer with much higher *cis*-1,4 units content (*cis*-1,4 > 97%). It was found that the complex catalyst system exhibited not only higher catalytic activity, but could also effectively control the stereo-structure of polybutadiene component within whole copolymer chains. The significant discovery was that the molar ratio of Li and B reagents could markedly affect the conversion of butadiene monomer to determine the copolymer’s stereo-structure and its molecular weight distribution. The optimal molar ratio for Li and B reagents was fixed to [Li]/[Ni] = 5 and [B]/[Li] = 1 based on the experimental data. Besides, the copolymer’s block ratio could also be controlled via controlling the monomer’s feed ratio. DSC measurements indicated that the copolymer synthesized via Ni/PSLi/B possesses an excellent lower temperature behavior, as compared to the copolymer obtained via *n*-butyllithium catalyst system.

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
