# Peer review of "A New Synthesis Strategy on Styrene-Butadiene Di-Block Copolymer Containing High cis-1,4 Unit Via Transfer of Anionic to Coordination Polymerization"

_polymers, 2019, doi:10.3390/polym11020195_

Round 1

Reviewer 1 Report

This paper by Fan et al. presents block copolymerization of styrene with 1,3-butadiene using anionic polymerization and coordination polymerization using a Ni catalyst for the second monomer polymerization.  The manuscript is not suited for publication in its present form.  I recommend the authors to address the following issues.

1) Copolymer with high selectivity for 1,4-cis structure for the butadiene segment is already reported as mentioned by the authors (refs 12 and 13).  Only the price of a catalyst and a cocatalyst does not rationalize its publication because catalysts with higher productivity or with better control of the molecular weight.  The authors need to present other advantage over or difference from the catalyst composed of the lanthanide compounds reported so far.   

2) Chemical formulae of the polymer is wrong throughout the paper.  Styrene units should be shown as –(-CHPh-CH2)- rather than –(-CHPh-)-.  Units of a block copolymer should be presented by brackets while those of a random copolymer should be separated by slash.  Thus, the polymer can be shown as follows (or the drawing attched).  This issue is so serious in the manuscript for a scientific journal.

3) The authors should mention homopolymerization of 1,3-butadiene catalyzed by Nickel naphtenate to compare it with results of the copolymerization, such as productivity, molecular weight of the product.  If it was already reported, proper citation of the paper or patent is needed.

4) Detailed experimental conditions, including amounts of the monomers, catalyst, and co-catalyst, are scattered in captions of Figures 1, 2, and 4, and in footnote of Table 1. Experimental section, however, mentions more simple and general experimental procedure for the reaction.  The authors should describe the established in details for the anionic block copolymerization of styrene and 1,3-butadiene and the successive copolymerization using BuLi initiator and Ni catalyst, and cite them in the figure captions. 

5) The copolymerization appears to be free from chain transfer.  Molecular weight of the copolymer (8.3 x 104, entry 1 Table 1), however, is much smaller than that calculated from the ratio of the monomers to the Ni catalyst (15 x 104).    

Author Response

Response to Reviewer 1 Comments

Point 1: Copolymer with high selectivity for 1,4-cis structure for the butadiene segment is already reported as mentioned by the authors (refs 12 and 13).  Only the price of a catalyst and a cocatalyst does not rationalize its publication because catalysts with higher productivity or with better control of the molecular weight.  The authors need to present other advantage over or difference from the catalyst composed of the lanthanide compounds reported so far.

Response 1:

Thanks for your suggestion. Based on the current theory that high cis-1,4 structure content can significantly improve the low temperature resistance and mechanical properties performance of styrene-butadiene block di-block (PS-b-PB) or tri-block (SBS) copolymeras shown in Line 37-39 in the revised manuscript, we aimed to develop a simple an inexpensive and efficient synthesis process for styrene-butadiene block copolymers with high cis-1,4 structure content. In this manuscript, the nickel naphthenate was used as the main catalyst, which is a simple, cheap and easily obtained compound. Since the synthesis theory study of polybutadiene with high cis-1,4 content via nickel naphthenate/ triisobutylaluminium/boron trifluoride etherate catalyst system is very mature, [1-5] we had innovatively substituted triisobutylaluminium with macromolecular alkylating initiator (butyl lithium-initiated styrene) for the synthesis of block copolymers PS-b-PB. The catalytic system(Ni/PSLi/B) has high catalytic activity for butadiene under mild reaction conditions, which can effectively control the molecular weight of the product, and the most notable is high cis-1, 4 selectivity (cis-1,4 >97%).

The two cationic alkyl rare-earth metal systems reported by Cui etal. [6] (ref [10] in the revised manuscript) and Hou etal. [7-9] (ref [22] in the revised manuscript) mentioned in the orginal manuscript can indeed achieve the copolymerization of styrene and butadiene efficiently, to obtain the di-block polymers with high cis-1,4 structure content. Theses two synthesis method of PS-b-PB also have potential application prospects, but we believe that our nickel Ni/PSLi/B still has some advantage in industrial application.

In the first article reported by Cui et al., [6] the catalyst linked-half-sandwich-lutetium–bis(allyl) complex [C5Me4-C5H4N)Lu(η3-C3H5)2] was synthesized by [Lu(η3-C3H5)3(diox)] (diox=1,4-dioxane) and C5Me4-C5H4N. Hovever, these two compounds are both not commercially available, which were synthesized by a rigour and complicated process. Furthermore, the raw material used in above synthesis and the co-catalyst [Ph3C][B(C6F5)4] in the polymerization are all expensive. These factors are all not conducive to the industrialization application of this catalyst system. In addition, it can be seen from the experimental results (in the Table 2, ref [10] in the revised manuscript), the butadiene cis-1,4 is about 95%, which is slightly lower than the PS-b-PB obtained by the Ni/PSLi/B system (cis-1, 4>97 %).

The another type of cationic alkyl rare-earth metal systems, Half-Sandwich samarium or scandium catalyst were reported by Hou etal., [7-9] which introduced that the styrene-butadiene di-block copolymer was prepared via (C5Me5)2Sm(μ-Me2)AlMe2/[Ph3C][B(C6F5)4] catalyst system. The cis-1,4 structure content in the obtained PS-b-PB block copolymer was 98.8% and the styrene content was 5.5%. But the polymerization activity of styrene was slightly low and the reaction needed to carried out at -20 °C. In addition, the Half-Sandwich scandium or samarium catalyst were also synthesized via a rigour and complicated synthesis process, and co-catalyst [Ph3C][B(C6F5)4] needed in the polymerization is expensive.

According to the reviewer's suggestion, in order to further illustrate the advantage of Ni/PSLi/B in the synthesis of PS-b-PB, we correct the senetences (Line 57-61 in the original manusrcript) “Cui and Hou were respectively synthesized styrene-butadiene di-block copolymers with high cis-1,4 unit content (cis-1,4 >97%) using a cationic alkyl rare-earth metal catalyst. Although the conversion rate was higher (>99%) in each steps, the cost of catalyst and co-catalyst [Ph3C][B(C6F5)4] seriously limited their industrial-scaled synthesis and application” to “Cui et al. synthesized PS-b-PB with high cis-1,4 unit content (cis-1,4 ~95%) by utilizing catalyst [C5Me4-C5H4N)Lu(η3-C3H5)2]. Hou et al. also have synthesized the PS-b-PB with 99% cis-1,4 unit content using a Half-Sandwich samarium catalyst. However, the complicated and expensive preparation process of these half-sandwich rare earth catalyst and the cost of the co-catalyst [Ph3C][B(C6F5)4] seriously limited their industrial-scaled synthesis and application.” (Line 62-66 in the revised manuscript)

In addition, in order to further illustrate the cationic alkyl rare-earth metal systems, we have supplemented the reference “Random- and Block-Copolymerization of 1,3-Butadiene with Styrene Based on the Stereospecific Living System: (C5Me5)2Sm(μ-Me2)AlMe2/[Ph3C][B(C6F5)4]” in the revised manuscript.(referene [23] in the revised manuscript)

Point 2: Chemical formulae of the polymer is wrong throughout the paper.  Styrene units should be shown as –(-CHPh-CH2)- rather than –(-CHPh-)-.  Units of a block copolymer should be presented by brackets while those of a random copolymer should be separated by slash.  Thus, the polymer can be shown as follows (or the drawing attched).  This issue is so serious in the manuscript for a scientific journal.

Response 2:

First, I really apologize for my mistakes. We have corrected the structural formula of the PS-b-PB copolymers in the scheme 1 and Fig.3. (Line 75 and Line 185 in the revised manuscript)

Scheme 1. Synthesis routes of the PS-b-PB via the strategy of "anion to coordination" transfer mechanism.

Fig.3. 13C NMR spectrum of PS-b-PB obtained via Ni/PSLi/B catalyst system. (PS-b-PB-4)

Point 3: The authors should mention homopolymerization of 1,3-butadiene catalyzed by Nickel naphtenate to compare it with results of the copolymerization, such as productivity, molecular weight of the product.  If it was already reported, proper citation of the paper or patent is needed.

Response 3:

Thanks for your suggestion. In our experiment, indeed, the homopolymerization of butadiene was achieved via catalyst system nickel naphthenate/n-butyllithium/boron trifluoride ether, and the cis-1,4 content in the obtained polybutadiene was 96.7%, the molecular weight (Mn) was 49,000, and the conversion was 87.6%. This result had been introduced simply in Line 255-258 of the original manuscript. According to the reviewer's suggestion, in order to further illustrate this comparison result, we decide to add the specific data in Table 1. (in the revised manuscript)

Table 1 The PS-b-PBs and homopolybutadiene structure via different catalyst system.

samples

Convd

%

St conte

mol %

Mnf*104

g/mol

Mw/Mnf

cis-1,4g

%

trans-1,4g

%

1,2-g

%

PS-b-PB-1a

91.8

25.8

8.3

1.29

97.1

0.6

2.3

PS-b-PB-2b

67.2

33.9

6.2

1.16

48.9

39.8

11.3

PB-1c

87.6

0

4.9

1.85

96.7

0.8

2.5

a. Di-block copolymerization of Styrene and Butadiene via Ni/PSLi/B catalyst system. b. Di-block copolymerization of styrene and butadiene via n-butyllithium catalyst system. c. Homopolymerization of butadiene via Ni/Li/B catalyst system. d. The conversion of butadiene monomer in the second step. e. Detercmined by the 1H NMR spectrum. f. Measured by SEC-MALLS. g. Detercmined by the 1H NMR and 13C NMR spectrum.

In addition, in order to further illustrate this comparison, we corrected sentences “In addition, during from anionic to coordination transfer polymerization, the monomer conversion data of butadiene in its homopolymer were obtained via aging nickel isophthalate (Ni), n-butyllithium(Li) and boron trifluoride ether (B) at 50°C for 15 minutes.” (Line 199-201 in the original manuscript) to “In addition, in order to show the conversion process from anion to coordination conversion process during the polymerization process, the catalyst system Ni/Li/B, obtained by aging the nickel naphthenate (Ni) with butyllithium(Li) and boron trifluoride ether (B), was studied to directly initiate the homopolymerization of butadiene.” (Line 255-258 in the revised manuscript)

Dixon et al. [10] have reported the synthesis of high cis-1,4 homo-polybutadiene (cis-1,4~96%) by a similar catalyst system nickel diisopropyl salicylate/n-butyllithium/boron trifluoride ether, and the conversion of butadiene was over than 90%. We have cited this article in the revised manuscript. (reference [23]). And the nickel naphtenate /butyllithium/boron trifluoride ether is a mature catalyst system and widely used in Ni-BR industry, the cis-1,4 unit content of these polybutadiene product are over than 97%, and the conversion rate are also generally higher than 85%. [1-5]

Point 4: Detailed experimental conditions, including amounts of the monomers, catalyst, and co-catalyst, are scattered in captions of Figures 1, 2, and 4, and in footnote of Table 1. Experimental section, however, mentions more simple and general experimental procedure for the reaction. The authors should describe the established in details for the anionic block copolymerization of styrene and 1,3-butadiene and the successive copolymerization using BuLi initiator and Ni catalyst, and cite them in the figure captions.

Response 4:

Thanks for your suggestion. Firstly, we have further improved the description of synthesis process of the PS-b-PB parpared by Li/PSLi/B system as follows “A detailed polymerization procedure (PS-b-PB-1, provided in Table 1) was described as a typical example. First, styrene polymerization was conducted in Schlenk tube with a rubber septum in following steps: styrene solution (10 mmol, 2M in cyclohexane, [St]/[Li]=200), Li (0.05 mmol, 2.5 M in n-hexane, [Li]/[Ni]=5), polymerization was carried out at 50 for 1 hour, and viscous orange liquid was obtained. When the Ni solution (0.01 mmol, 0.025M in cyclohexane) was added, the system became deep red by stirring at 50. After 15 min, B (0.05 mmol, [B]/[Li]=1) was added, and aged with stirring at 50 for 15 min, and a dark brown-colored catalyst Ni/PSLi/B, was successfully obtained. Then butadiene (10 mmol, 1.9 M solution in hexane, [Bd]/[Ni]=1000) and aforementioned prepared catalyst Ni/PSLi/B solution was injected into the Schlenk tube. This polymerization was carried out at 50 for 3 hours and then quenched by adding 2 ml ethanol containing antiager 264 (1%, w.t) as a stabilizer. The product was precipitated in methanol and repeatedly washed with ethanol, followed by extraction with methyl ethyl ketone and n-hexane, respectively. Then, the product was dried under vacuum at 40 for getting a constant weight. Finally, a white solid had been obtained. A series of other copolymers, such as PS-b-PB-1, PS-b-PB-2, PS-b-PB-3, PS-b-PB-5, PS-b-PB-6 (as shown in Table 2 for details), were synthesized by changing the amount of butadiene added in the second step of the polymerization with above synthesis process.” (Line 103-118 in the revised manuscript)

The detailed experimental description of PS-b-PB synthesized by n-butyllithium catalyst system was supplemented as follows “2.3 The synthesis of PS-b-PB via n-butyllithium catalyst systems. All syntheses were conducted in a dry argon atmosphere. The specific experimental process (PS-b-PB-a, provided in Table 1) is as follows. First, styrene polymerization was conducted in Schlenk tube with a rubber septum in following steps: styrene solution (15 mmol, 2M in cyclohexane, [St]/[Li]=300), Li (0.05mmol, 2.5 M in n-hexane), polymer synthesis was carried out at 50 for 1 hour, and viscous orange liquid was obtained. Then butadiene (50 mmol, 1.9 M solution in hexane, [Bd]/[Li]=1000) was injected into the Schlenk tube. This polymerization was carried out at 50 for 12 hours and then quenched by adding 2 ml ethanol containing antiager 264 (1 wt %) as a stabilizer. The product was precipitated in methanol and repeatedly washed with ethanol, followed by extraction with methyl ethyl ketone and n-hexane, respectively. Then, the product was dried under vacuum at 40 for getting a constant weight. Finally, a white solid has been obtained. (Line 124-140 in the revised manuscript)

The detailed experimental description of homopolymerization of butadiene was supplemented as follows”2.4. Homopolymerization of butadiene. All syntheses were conducted in a dry argon atmosphere. The specific experimental process (PB-1, provided in Table 1) is as follows. First, the alkylation process of nickel naphthenate was conducted in Schlenk tube with a rubber septum in following steps: butadiene (0.25 mmol, 1.9M in n-hexane, [Bd]/[Li]=5), Li (0.05mmol, 2.5 M in n-hexane, [Li]/[Ni]=5), Ni (0.01 mmol, 0.025M in cyclohexane). The reaction was carried out at 50 for 15 min, then the B (0.05 mmol, [B]/[Li]=1) was added by a micro-injector and aged with stirring at 50 for 15 min, and a dark brown-colored catalyst Ni/Li/B, was successfully obtained. Then butadiene (10 mmol, 1.9 M solution in hexane, [Bd]/[Ni]=1000) was injected into the Schlenk tube. This polymer polyerization was carried out at 50 for 3 hours and then quenched by adding 2 ml ethanol containing containing antiager 264 (1 wt %). The product was precipitated in methanol and repeatedly washed with ethanol. Then, the product was dried under vacuum at 40 for getting a constant weight. Finally, a white solid has been obtained.” (Line 141-152 in the revised manuscript)

In addition, in order to avoid repetition, the caption of Fig.1, Fig.2, Fig.3, Fig.4, Fig.5 and Fig.6 in the original manuscript and the footnote of Table 1 and Table 2 were revised, and please see the revised manuscript for details.

Point 5: The copolymerization appears to be free from chain transfer.  Molecular weight of the copolymer (8.3 x 104, entry 1 Table 1), however, is much smaller than that calculated from the ratio of the monomers to the Ni catalyst (15 x 104). 

Response 5:

This suggestion is very useful to us. It should be pointed out that a certain percentage of macromolecular alkylating initiator (PSLi) could participate in the alkylation process of nickel naphthenate. This mechanism may actually affect the efficient utilization of PSLi during copolymerization. The result of the mechanism is that the molecular weight (Mn) of the di-block copolymer is lower than the calculated value. In addition, in the second step of the copolymerization (the polymerization of butadiene), the conversion of butadiene is about 90%, which also makes the difference between the molecular weight (Mn) and the calculated value of the di-block copolymer.

Reference

[1] Taube R.; Windisch H.; Maiwald S. The catalysis of the stereospecific butadiene polymerization by Allyl Nickel and Allyl Lanthanide complexes - A mechanistic comparison, Macromol. Symp., 1995,89,393 - 409.

[2] Throckmorton M.C.; Farson F. S. An HF-Nickel-R3Al Catalyst System for Producing High Cis-1,4-Polybutadiene. Rubber Chemistry and Technology 1972, 45, 268–277.

[3] Matsumoto T., Furukawa, J.; Morimura H. Polymerization of butadiene catalyzed by the reaction product of π-allyl nickel halide with organic peroxide or oxygen. Journal of Polymer Science Part B: Polymer Letters, 1969, 541–546. doi:10.1002/pol.1969.110070710

[4] Porri, L., Natta, G., Gallazzi, M. C. Stereospecific polymerization of butadiene by catalysts prepared fromπ-allyl nickel halides. Journal of Polymer Science Part C: Polymer Symposia, 1967,16, 2525–2537.

[5] D.B. Chen, G. Xu, X.M. Tang, synthesis of cis-1,4 polybutadiene with nickel catalyst, VII Utraviolet and visible speetra study of interaction between component in nickel catalyst system. Acta Polymerica Sinica 1987, 2 ,99-105.

[6] Jian, Z.B.; Tang, S.; Cui, D.M. A Lutetium Allyl Complex That Bears a Pyridyl-Functionalized Cyclopentadienyl Ligand: Dual Catalysis on Highly Syndiospecific and cis-1,4-Selective (Co)Polymerizations of Styrene and Butadiene. Chem-Eur. J. 2010, 16, 14007-14015.

[7] Li, P.; Zhang, K.Y.; Nishiura, M.; Hou, Z.M.; Chain-Shuttling Polymerization at Two Different Scandium Sites: Regio- and Stereospecific “One-Pot” Block Copolymerization of Styrene, Isoprene, and Butadiene, Angew. Chem. Int. Edi. 2011, 50, 12012-12015.

[8] Guo, F.; Meng, R.; Li, Y.; Hou, Z. (2015). Highly cis -1,4-selective terpolymerization of 1,3-butadiene and isoprene with styrene by a C5 H5 -ligated scandium catalyst. Polymer 2015, 76, 159–167.

[9] Kaita S.; Hou Z.M.; Wakatsuki Y. Random- and Block-Copolymerization of 1,3-Butadiene with Styrene Based on the Stereospecific Living System: (C5Me5)2Sm(μ-Me2)AlMe2/[Ph3C][B(C6F5)4][1].  Macromolecules 2001, 34, 1539-1541.

[10] Dixon C.; Duck E.W.; Grieve D.P.; Jenkins D. K.; Thornber M. N. High cis-1,4 polybutadiene synthesis-The catalystsystem nickel diisopropyl salicylate, boron trifluoride ether, butyllithium. Eur. Polym. J. 1970, 6, 1359-137.

Reviewer 2 Report

The authors described a new method to synthesize polystyrene-block-polybutadiene by using sequence anionic and coordination polymerizations. The resulting polybutadiene block contains high cis 1,4 addition products which are beneficial for many applications. The key idea is the mechanism transfer in the synthesis. The method could be very useful in the industry. However, the authors must improve the quality of the presentation before further consideration. 

There are many grammar mistakes in the manuscript. 

In the experimental section, the exact procedures of the experiments need to be described so that the experiments could be reproduced.

All reaction products should be listed and numbered with their reactant ratio, molecular weight, and molecular weight distribution provided, so the readers could understand which product is being described in the figures and tables. 

Author Response

Response to Reviewer 2 Comments

Point 1: There are many grammar mistakes in the manuscript.

Response 1:

Thanks for your suggestion. We have got help from a native English speaker to improve our English express in this manuscript. And this article has been carefully checked repeatedly, the grammatical and wording errors you mentioned had been corrected and marked using "Track Changes" function in the revised manuscript.

Point 2: In the experimental section, the exact procedures of the experiments need to be described so that the experiments could be reproduced.

Response 2:

First, I really apologize for the inconvenience for you caused by our not exhaustive description of the experimental procedures.

Firstly, we have further improved the description of synthesis process of the PS-b-PB parpared by Li/PSLi/B system as follows “A detailed polymerization procedure (PS-b-PB-1, provided in Table 1) was described as a typical example. First, styrene polymerization was conducted in Schlenk tube with a rubber septum in following steps: styrene solution (10 mmol, 2M in cyclohexane, [St]/[Li]=200), Li (0.05 mmol, 2.5 M in n-hexane, [Li]/[Ni]=5), polymerization was carried out at 50 for 1 hour, and viscous orange liquid was obtained. When the Ni solution (0.01 mmol, 0.025M in cyclohexane) was added, the system became deep red by stirring at 50. After 15 min, B (0.05 mmol, [B]/[Li]=1) was added, and aged with stirring at 50 for 15 min, and a dark brown-colored catalyst Ni/PSLi/B, was successfully obtained. Then butadiene (10 mmol, 1.9 M solution in hexane, [Bd]/[Ni]=1000) and aforementioned prepared catalyst Ni/PSLi/B solution was injected into the Schlenk tube. This polymerization was carried out at 50 for 3 hours and then quenched by adding 2 ml ethanol containing antiager 264 (1%, w.t) as a stabilizer. The product was precipitated in methanol and repeatedly washed with ethanol, followed by extraction with methyl ethyl ketone and n-hexane, respectively. Then, the product was dried under vacuum at 40 for getting a constant weight. Finally, a white solid had been obtained. A series of other copolymers, such as PS-b-PB-1, PS-b-PB-2, PS-b-PB-3, PS-b-PB-5, PS-b-PB-6 (as shown in Table 2 for details), were synthesized by changing the amount of butadiene added in the second step of the polymerization with above synthesis process.” (Line 103-118 in the revised manuscript)

Secondly, the experimental description of PS-b-PB synthesized by n-butyllithium catalyst system was supplemented (Line 124-140 in the revised manuscript) as follows “2.3 The synthesis of PS-b-PB via n-butyllithium catalyst systems. All syntheses were conducted in a dry argon atmosphere. The specific experimental process (PS-b-PB-2, provided in Table 1) is as follows. First, styrene polymerization was conducted in Schlenk tube with a rubber septum in following steps: styrene solution (15 mmol, 2M in cyclohexane, [St]/[Li]=300), Li (0.05mmol, 2.5 M in n-hexane, [Li]/[Ni]=5), polymer synthesis was carried out at 50 for 1 hour, and viscous orange liquid was obtained. Then butadiene (50 mmol, 1.9 M solution in hexane, [Bd]/[Li]=1000) was injected into the Schlenk tube. This polymerization was carried out at 50 for 12 hours and then quenched by adding 2 ml ethanol containing antiager 264 (1, wt %) as a stabilizer. The product was precipitated in methanol and repeatedly washed with ethanol, followed by extraction with methyl ethyl ketone and n-hexane, respectively. Then, the product was dried under vacuum at 40°C for getting a constant weight. Finally, a white solid has been obtained.”

Finally, the experimental description of homopomerization of butadiene via Ni/Li/B catalyst system was supplemented (Line 141-153 in the revised manuscript) as follow “2.4. Homopolymerization of butadiene. All syntheses were conducted in a dry argon atmosphere. The specific experimental process (PB-1, provided in Table 1) is as follows. “First, the alkylation process of nickel naphthenate was conducted in Schlenk tube with a rubber septum in following steps: Ni (0.01 mmol, 0.025M in cyclohexane), Li (0.05mmol, 2.5 M in n-hexane, [Li]/[Ni]=5). The reaction was carried out at 50 for 15 min, then the B (0.05 mmol, [B]/[Li]=1) was added by a micro-injector and aged with stirring at 50 for 15 min, and a dark brown-colored catalyst Ni/Li/B, was successfully obtained. Then butadiene (10 mmol, 1.9 M solution in hexane, [Bd]/[Ni]=1000) was injected into the Schlenk tube. This Apolymerization was carried out at 50 for 3 hours and then quenched by adding 2 ml ethanol containing antiager 264 (1 wt %). The product was precipitated in methanol and repeatedly washed with ethanol. Then, the product was dried under vacuum at 40 for getting a constant weight. Finally, a white solid has been obtained.

In addition, in order to avoid repetition, the caption of Fig.1, Fig.2, Fig.3, Fig.4, Fig.5 and Fig.6 in the original manuscript and the footnote of Table 1 and Table 2 were revised, and please see the revised manuscript for details.

Point 3: All reaction products should be listed and numbered with their reactant ratio, molecular weight, and molecular weight distribution provided, so the readers could understand which product is being described in the figures and tables.

Response 3:

This suggestion is very useful to us. We have numbered the polymers involved in the paper according to the reviewer’s proposal according to the catalyst system and the ratio of monomer feed, such as “PS-b-PB-1” and so on. (as shown in the Table 1 and Table 2 of the revised manuscript). The description of the figure caption and the table footnote and the expression in the main text has been modified. And I really apologize for the inconvenience for you caused by this problem.

Round 2

Reviewer 2 Report

The quality of the presentation still needs to be improved after the revision. The authors should review the manuscript very carefully. There are many mistakes, such as in Line 15 “a complex catalyst sys was tem”; Line 46 and 47 repeated “with”; Line 48 “as higher as” should be “as high as”; Line 98 “Synthesis PS-b-PB” should be “Synthesis of PS-b-PB”; Line 101 “The n-butyllithium directly purchased was used” should be “The n-butyllithium was used directly as purchased”; Line 116 “a white solid had been obtained” should be “a white solid was obtained”. Those are just a few examples.

The positions of the brackets in Scheme 1 and Fig 3 are incorrect. 

 What were the monomer conversion, molecular weight and molecular weight distribution of the PS after the first step polymerization? 

Author Response

Point 1: The quality of the presentation still needs to be improved after the revision. The authors should review the manuscript very carefully. There are many mistakes, such as in Line 15 “a complex catalyst sys was tem”; Line 46 and 47 repeated “with”; Line 48 “as higher as” should be “as high as”; Line 98 “Synthesis PS-b-PB” should be “Synthesis of PS-b-PB”; Line 101 “The n-butyllithium directly purchased was used” should be “The n-butyllithium was used directly as purchased”; Line 116 “a white solid had been obtained” should be “a white solid was obtained”. Those are just a few examples.

Response 1:

Thanks very much for your suggestion. We have revised the errors that you have pointed out using "Track Changes" function with yellow highlights in the revised manuscript. In addition, this article has been checked repeatedly, and other grammatical and expression errors have also been corrected using "Track Changes" function with yellow highlights in the revised manuscript.

Point 2: The positions of the brackets in Scheme 1 and Fig. 3 are incorrect.

Response 2:

First, we really apologize for our mistakes. We have corrected the structural formula of the PS-b-PB copolymers in the scheme 1 and Fig.3. (Line 76 and Line 189 in the revised manuscript)

 Scheme 1. Synthesis routes of the PS-b-PB via the strategy of "anion to coordination" transfer mechanism.

Fig.3. 13C NMR spectrum of PS-b-PB obtained via Ni/PSLi/B catalyst system. (PS-b-PB-4)

Point 3: What were the monomer conversion, molecular weight and molecular weight distribution of the PS after the first step polymerization?

Response 3:

Please allow me take a detailed polymerization (PS-b-PB-4, provided in Table 1) as an example. Firstly, styrene (St, 10 mmol, 2 M in cyclohexane, [St]/[Li]=200) was initiated by n-butyllithium (Li, 0.05 mmol, 2.5 M in n-hexane) utilizing anionic polymerization at 50°C for 1 hour. We have terminated the reaction in the first step of the polymerization and characterized the obtained polystyrene. As a result, the monomer conversion of styrene was 99.1%, and the molecular weight (Mn) and molecular weight distribution (Mw/Mn) of the obtained PS were 2.2 x104 kg/mol and 1.05, respectively.
